# Exploring Financial Challenges and University Support Systems for Student Financial Well-Being: A Scoping Review

**DOI:** 10.3390/ijerph22030356

**Published:** 2025-02-27

**Authors:** Joanna Russell, Kylie Austin, Karen E. Charlton, Ezinne O. Igwe, Katherine Kent, Kelly Lambert, Gabrielle O’Flynn, Yasmine Probst, Karen Walton, Anne T. McMahon

**Affiliations:** 1School of Health and Society, Faculty of the Arts, Social Sciences and Humanities, University of Wollongong, Wollongong, NSW 2522, Australia; jrussell@uow.edu.au; 2Student, Equity, and Success, University of Wollongong, Wollongong, NSW 2522, Australia; kaustin@uow.edu.au; 3School of Medical, Indigenous, and Health Sciences, Faculty of Science, Medicine, and Health, University of Wollongong, Wollongong, NSW 2522, Australia; karenc@uow.edu.au (K.E.C.); ekorie@uow.edu.au (E.O.I.); katherinek@uow.edu.au (K.K.); klambert@uow.edu.au (K.L.); yasmine@uow.edu.au (Y.P.); kwalton@uow.edu.au (K.W.); 4School of Education, Faculty of the Arts, Social Sciences and Humanities, University of Wollongong, Wollongong, NSW 2522, Australia; gabriell@uow.edu.au

**Keywords:** financial challenges, food insecurity, student well-being, university students, support services

## Abstract

Background/Objectives: Financial stress among university students has been widely documented, impacting academic performance, mental health, and overall well-being. This scoping review explores and synthesizes the existing evidence on the extent and impact of financial challenges experienced by university students in Australia and New Zealand and examines approaches implemented by universities in these countries to address these challenges. Methods: The Arksey and O’Malley framework was utilized for comprehensiveness, structure, and reproducibility. Four scientific databases (Scopus, ProQuest, Web of Science, and Informit) were searched until 30 June 2024, and 3542 articles were identified. Following extensive screening, 19 studies were included. The studies were summarized using a narrative synthesis approach. Results: This review suggests that financial stress continues to be experienced by Australian/New Zealand university students. Some studies indicate that over half (8–68%) of students face significant financial issues and 96% of students report high emotional stress. Some groups are more vulnerable than others. Notably, students from low socio-economic status (SES) backgrounds and international students have increased vulnerability due to factors such as inadequate financial support and limited access to job opportunities and support services. Support services available for students included emergency grants, food pantries (including international students), and community gardens but with limited impact in addressing underlying financial hardships. Conclusion: This review highlights the persistent financial challenges faced by vulnerable university students in Australia and New Zealand. It calls for comprehensive strategies to enhance support services and address structural issues in government and institutional policies. Addressing these needs will enable improved student academic success and mental and physical well-being in these vulnerable groups.

## 1. Introduction

The financial landscape for university students has become increasingly complex as the global economy undergoes seismic shifts [1]. With rising tuition fees and costs of living, in addition to the added pressures of economic instability, students face unprecedented financial burdens [2]. These challenges are not only affecting their academic performance but also their overall well-being [3]. Although the global financial crisis (GFC) in 2007–2009 affected all facets of society, university students were among the most vulnerable groups affected [4,5]. One study conducted in Greece found that financial challenges significantly affected students’ overall quality of life and their predisposition toward pursuing qualifications beyond a Bachelor’s degree [6]. From an Australian perspective, available evidence indicates that the GFC significantly affected the overall well-being of adults aged 19–22 years [7]. In 2006, a national survey of Australian students found that approximately 13% of students went without necessities, such as food and medical attention, because of financial challenges [8]. In 2012, this number had increased by 30%, with the same proportion of students reporting that they missed classes regularly due to work commitments [9]. These impacts were more evident for students from regional areas, with students from a low socio-economic background studying at regional universities doubly affected [10]. This has been attributed to a complex interplay of factors, including the costs of study materials, travel to university, the usual expenses of living, and sometimes supporting a family, which contribute to the financial burden and challenges experienced [10]. Although there was a slight improvement in 2017 in the overall financial circumstances of Australian students (14% of students foregoing necessities [11]), the financial challenges experienced by other more vulnerable groups were highlighted. In addition to this, the economic and social disruptions associated with the COVID-19 pandemic (2020–2022) likely exacerbated these financial challenges, particularly for already vulnerable student groups. Job losses, reduced work hours, and the increased cost of living have intensified the financial pressures on students. For example, a 2024 study [12] conducted on Australian university students reported that financial challenges are compounded when students are undertaking a professional placement as part of their degree. The financial stress experienced is reflected in high levels of food insecurity (the absence of access to and the availability of enough healthy food), which was experienced by an alarming 78% of students surveyed while on their most recent placement. Concerningly, 36% of students reported facing *moderate* food insecurity—characterized by eating food of lower quality, variety, or desirability—and 30% experienced *severe* food insecurity, involving reduced food intake, regular meal skipping, or periods of hunger [12]. Beyond food insecurity, there is a limited comprehensive analysis of their specific impacts on vulnerable groups, especially following the global pandemic.

Given the well-documented financial challenges faced by university students due to broader economic conditions—such as changes in government education funding, rising living costs, and limited employment opportunities—and their impact on well-being and academic performance, it is crucial to examine how universities respond. This scoping review goes beyond summarizing financial impacts by focusing on synthesizing the services and support strategies implemented by universities in Australia and New Zealand to address these challenges.

## 2. Materials and Methods

### 2.1. Study Design

A scoping review was conducted following the methodological framework outlined by Arksey and O’Malley (2005) [13]. The framework ensures a structured, comprehensive, and reproducible approach to mapping the existing literature on financial challenges and support systems for university students. The Preferred Reporting Items for Systematic Reviews and Meta-Analyses (PRISMA) Extension for Scoping Reviews checklist [14] was also utilized (Figure 1) in the reporting of the review, and the protocol was registered with the Open Science Framework (OSF) Registries (Reg. No: OSF.IO/BW7FQ) on 23 May 2024.

### 2.2. Study Questions

The “PCC” (population, concept, and context) mnemonic was utilized to identify the research questions guiding the scoping review to enable clarity for developing the search strategy and inclusion criteria [15].

Primary question: *What are the financial challenges and impacts experienced by university students in Australia and New Zealand?*

Secondary question: *What provisions, services, or practices are available to students facing financial hardships at universities in Australia and New Zealand?*

### 2.3. Search Strategy

Four scientific databases—Scopus, ProQuest, Web of Science, and Informit—were searched for relevant studies up to 30 June 2024. The search terms and keywords included “financial challenges”, “university students”, “higher education”, “financial aid”, and “support services”. Boolean operators and truncation were used to refine the search (Appendix C). A full search strategy was subsequently developed in consultation with a research librarian. An example of the search strategy in ProQuest is shown in the Appendix A.

### 2.4. Eligibility Criteria

Studies were included in the current scoping review if they met the following criteria:Outcome Focus: The study outcome reported on financial challenges faced by university students. This included studies that explored outcomes such as financial distress; financial concerns; financial worry; food insecurity; and housing affordability and availability. Food insecurity and housing affordability/availability were included because they are widely recognized in the literature as key indicators of financial hardship among university students [16].Population: The study population consisted of enrolled university students in Australia/New Zealand, including domestic and international students from undergraduate and postgraduate programs.Accessibility: All articles reviewed had to be accessible as a full-text record, ensuring that researchers could access and review the content.Language: Studies were required to be reported in English. This criterion was necessary due to the research team’s lack of access to translations for other languages.

Studies were excluded from the review based on the following criteria:Study design: Opinion pieces, commentaries, editorials, or non-empirical research that did not report original data or findings.Publication Type: Non-academic publications, such as blogs, magazines, or news articles that did not undergo a peer-review process.

### 2.5. Study Selection

A total of 3541 articles were identified from the initial search and imported into Covidence (Veritas Health Innovation, Melbourne, Australia) for screening. Covidence is a web-based collaboration platform that streamlines the production of systematic and other literature reviews [17]. After removing duplicates, three reviewers independently screened all titles and abstracts. Full texts of eligible studies were retrieved and screened by at least two reviewers per article. For the studies that did not meet the inclusion criteria in the full-text screening, the reasons for exclusion were recorded by both reviewers. Any discrepancies were addressed through discussion or consultation with a third reviewer.

### 2.6. Data Extraction and Synthesis

The data were extracted using a standardized form, capturing details on authors and year of publication, study objectives, methodologies, sample characteristics, key findings, types of financial challenges, support services provided, and barriers to accessing these services. The first reviewer completed the data extraction for all included studies and another five members of the research team extracted data from a subsample (~20%) independently. The results from all reviewers were compared and discussed to achieve accuracy and consistency. All discrepancies were addressed in a group discussion for consensus.

All study types (quantitative and qualitative studies) were extracted consistently using a predefined template. The main objectives and outcomes were described with any recommendations from the study reported. A descriptive summary of all included studies is presented in Table 1 and the results from all of the included studies were synthesized regarding the challenges facing university students in Australia and New Zealand.

## 3. Results

This review was reported according to the procedures and requirements described in the PRISMA (Preferred Reporting Items for Systematic Reviews and Meta-Analyses) extension for scoping reviews [14], ensuring transparency in the identification, screening, and inclusion of studies. The final database search identified 3541 articles (Figure 1). Following the removal of duplicates, titles and abstracts of 3340 studies were screened. After exclusion from the first screening, 23 relevant full-text articles studies were screened. Finally, 19 studies were eligible for inclusion. The reasons for excluding screened full-text articles are provided in Figure 1 and the Appendix A.

### 3.1. Study Populations

Twelve out of the nineteen articles included were based on results from individual universities in Australia or New Zealand [18,19,20,21,22,23,24,25,26,27,28,29], while the other seven pooled sample populations from several different universities and international campuses [9,30,31,32,33,34,35]. The characteristics of all included studies are shown in Table 1. A few of the studies limited their sample populations to undergraduate students, while most of the studies included all enrolled students. Two studies focused only on international students [21,32], while eight other studies included international student status as a covariate in their data analysis [9,25,26,28,29,31,33,35].

### 3.2. Types of Research

Fifteen studies had a cross-sectional study design; one study utilized a qualitative approach [18] and three studies utilized mixed methods approaches [30,32,35]. One study collected data on participants over two time periods [21].

### 3.3. Study Outcomes

Seven studies examined food security among university students, while four focused primarily on the financial challenges faced by students. Other studies investigated a combination of these topics, including food security, financial hardship, and their associated impacts, such as housing insecurity, mental health, and academic performance. The severity of challenges experienced by students ranged from moderate to high with identified factors contributing to challenges: low SES, underserved groups (Indigenous and international students), household situation/living away from home, unemployment/on benefits/employment status, homesickness, COVID-19 and the loss of employment.

Financial Challenges: Financial difficulties were consistently identified as a significant challenge for university students, both domestic and international [9,18,21,29,30,33,35]. The prevalence of financial hardship varied (8–68%), with one study in Australia [35] reporting that 58% of domestic undergraduates and 50% of international students faced financial difficulties where their financial situation was often a source of worry for them. Many students reported forgoing necessities, such as food and adequate housing to manage their studies. Bexley et al. (2013) [9] highlighted that 68% of full-time students in Australia experienced financial difficulty, further exacerbating their academic pressures.

Food Insecurity: Food insecurity was a common outcome, with varying levels of severity reported. Thielking et al. (2019) [29] found that 26% of students experienced moderate to severe food insecurity, while Gallegos et al. (2014) [19] noted moderate food insecurity in 26% of students. In a more recent study by Kent et al. (2023) [22], 54% of students were classified as food-insecure (23% moderate and 18% severe), which was associated with poorer diet quality among students. These outcomes were closely linked to financial instability, highlighting the inability of students to access sufficient and nutritious food, leading to both physical and emotional strain.

Mental Health and Well-being: Financial stress and food insecurity were closely tied to adverse mental health outcomes. For example, Ke et al. (2023) [21] reported moderate to high levels of depression and anxiety (22–57%) among Chinese international students, driven by financial hardships and social isolation. Moscaritolo et al. (2022) [33] reported 96% of students as having high levels of emotional stress, and 74% faced significant financial challenges, contributing to deteriorating mental health. The impact of the COVID-19 pandemic further exacerbated these conditions, especially amongst international students who were particularly vulnerable due to their socio-economic circumstances and limited access to support services.

Availability of Support Services: Eight studies explored available support services for university students, including food assistance (e.g., food banks, free meals), financial aid (e.g., emergency grants, subsidized meals), social support (e.g., mental health services, counselling), and academic initiatives (e.g., workshops, sustainable food programs) [9,20,21,22,26,28,32,33]. However, there were barriers to accessing these services due to a lack of awareness, stigma, and accessibility challenges, particularly for international students [9,24,26,32]. While some interventions, like social support during the COVID-19 pandemic, proved highly effective in reducing mental health issues, others, such as food assistance programs, were underutilized or insufficient [20,21,26,32]. These findings emphasize the need for the better communication of available services, expanded eligibility, and culturally sensitive approaches to improve access and impact.

*Effectiveness of Support Services:* A total of 12 studies indicated that while some universities had implemented support services [9,18,19,21,22,23,25,26,28,29,32,33], there was little evidence of these services being highly effective. Five studies did not report on the effectiveness of these support services [9,22,25,28,33]; others found that due to underutilization, the effectiveness of these support services could not be explored [19,29]. Other forms of support, such as financial aid or emergency grants, were reported but were often found lacking in their capacity to fully alleviate students’ financial burdens. For instance, Kent et al. (2023) [22] described on-campus food pantries and community gardens as being helpful but insufficient to address the broader issue of food insecurity. Similarly, another study [26] observed that 41% of students were satisfied with the food available on campus. While Ke et al. (2023) [21] found that social support significantly reduced the risk of major depression among Chinese international students, this was not the case for all students, as not everyone had access to robust social support systems.

*Barriers to Accessing Support:* A common outcome across multiple studies was the existence of barriers preventing students from accessing available support services. Moscaritolo et al. (2022) [33] pointed to cultural and language barriers, particularly for international students as major obstacles. Corney et al. (2024) [32] also highlighted the cost of accessing support services, alongside a lack of awareness and stigma associated with seeking help, as a significant factor limiting the effectiveness of existing services. Overall, the stigma associated with accessing support was highlighted across numerous articles [19,20,22,25,28,29,32,33,34,35]. Moreover, there were logistical challenges, such as the limited operating hours of food pantries, making it difficult for students to take full advantage of these services.

*Impact of the COVID-19 Pandemic:* The global pandemic had a profound impact on students’ financial and emotional well-being. Two articles [21,27] documented the loss of employment and increased isolation as key drivers of financial stress and mental health issues. Reduced work opportunities and increased costs related to remote learning further compounded the financial strain faced by students. Additionally, international students were disproportionately affected due to the lack of access to government benefits and heightened social isolation.

**Table 1 ijerph-22-00356-t001:** Characteristics of included studies on the financial challenges faced by university students in Australia.

Reference (Author, Year)	Population and Study Design	Study Objective/Key Concepts Explored	Type and Severity of Challenges ^1^	Factors Contributing to Challenges	Types and Effectiveness of Support Services	Barriers to Accessing Support	Recommendation and Conclusion
Universities Australia (2018) [35]	Population: Students (domestic and international) from 38 Universities Australia (UA) member universities (n = 18,584); 7.2% Indigenous Australians Study design/approach: Cross-sectional/quantitative and qualitative	To assess the financial circumstances of Australian university students, with a focus on income, expenditure, and financial hardship.	Type: Financial challenges; food insecurity. Severity: High (58% in undergraduate domestic students and 50% in international undergraduate students).	Low SES; Indigenous status; regional areas of residence.	Type: NR. Effectiveness: NR.	Lack of awareness of support services, limited eligibility for financial aid (especially for international students), and stigma around seeking assistance.	Recommendations include increasing financial aid for vulnerable student groups, improving work–study balance, and providing targeted support for Indigenous, low SES, and international students. Policy changes to increase income support were also suggested.
Gallegos, D., et al. (2014) [19]	Population: Students from the Business and Health Faculty at an Australian university (n = 810); 0.4% Indigenous Australians Study design/approach: Cross-sectional/quantitative	To investigate the extent and severity of food insecurity among tertiary students, as well as its association with sociodemographic and health factors.	Type: Food insecurity. Severity: Moderate food insecurity. (26%)	Household structure; housing tenure; household income; unemployment; perceived poor health; and dietary habits.	Type: Food relief strategies, such as university-sponsored food banks, were available, but only 5.6% of students had accessed them. Effectiveness: Effectiveness was limited due to underutilization.	Stigma associated with accessing food relief, lack of awareness about available support services, and social/cultural barriers.	Improving access to and the availability and affordability of food on campus needs to be given priority, with the development of innovative strategies that maintain human dignity.
Thielking et. al. (2019) [29]	Population: Students aged 18 to 25 years enrolled in a full or partial on-campus undergraduate or TAFE course at Swinburne University of Technology (n = 1231); 0.7% Indigenous Australians Study design/approach: Cross-sectional/quantitative	To explore the prevalence of issues related to poverty and how such issues impact student well-being, learning, and retention.	Type: Financial stress; psychological distress; quality of life and satisfaction with health; socio-economic status (SES); accommodation insecurity; food insecurity. Severity: Moderate to high (53%) level of financial stress; moderate to high (53%) level of psychological distress; most students were either very satisfied (17%) or satisfied with their health (44%); around 14% of students were classified as LSES; 15% of students had experienced homelessness; around 26% of the students were food-insecure.	Being on government benefits; disability; work status; international students; income; and level of financial stress.	Type: University services (e.g., food bank, free meals). Effectiveness: Services were underutilized; however, most students (61.4%) were willing to seek help in the future.	Stigma associated with using services, lack of awareness, cultural and financial barriers for international students.	Universities need to consider providing student health services, promoting financial well-being, providing employment opportunities, promoting nutrition well-being, and eliminating hunger, as well as providing safe and secure student accommodation.
Ke et. al. (2023) [21]	Population: Chinese international students studying at a Melbourne tertiary education institute. Wave 1 (n = 2514); Wave 2 (n = 434) Study design/approach: Longitudinal/quantitative	To examine the effect of the 2020 pandemic on the mental health of Chinese international students living in Australia and China and the protective effect of social support.	Type: Financial difficulty; depression and anxiety. Severity: Low financial difficulty (7.7%); moderate to high prevalence of depression and anxiety (22–57%).	COVID-19-related stressors such as lockdown, homesickness, racial discrimination, financial difficulties, and isolation were significant contributors.	Type: Social support (6-item Medical Outcomes Study Social Support Survey (MOS- SSS-6)). Effectiveness: High social support during the COVID-19 pandemic (Wave 2) was strongly associated with a decreased risk of reporting major depression in Wave 2. Students with high social support during the COVID-19 pandemic were 82% less likely to report major depression in Wave 2 compared to those students with low social support during the pandemic.	Limited access to community support, lack of informal social support networks, and isolation due to lockdowns and distance from family.	The major challenges identified existed before the pandemic and are likely to continue afterward. Universities and other educational institutions can use these findings to develop effective interventions to help Chinese international students cope.
Post et. al. (2021) [27]	Population: First-year pharmaceutical and medical science students (n = 126); racial breakdown not provided Study design/approach: Cross-sectional longitudinal survey/quantitative	To assess the impact of COVID-19 pandemic restrictions on student health, well-being, and social circumstances.	Type: Emotional well-being; motivation to study. Severity: Increased isolation (83%) and increased stress (67%); reduced motivation to study (89%).	Job loss due to COVID-19 restrictions; inability to find employment; social isolation; family stress; and transition to online learning.	Type: NR. Effectiveness: NR.	Barriers included lack of awareness about available support, stigma, and difficulty transitioning to online learning environments.	Several factors likely affect students’ ability to succeed in university during pandemic-like conditions, including decreased motivation, feelings of isolation from peers, challenges in understanding course content, technological issues, and unsuitable home environments.
Tani et. al. (2019) [34]	Population: Students enrolled in different programs at a higher education provider in New Zealand (n = 216); 48% European, 25% Asian, 20% Māori, 1.9% Pasifika, and other ethnicity (3.7%) Study design/approach: cross-sectional survey/quantitative	To explore the effect of external factors on students’ academic performance in higher education, with a focus on attendance and work–life balance.	Type: Students, grade point average (GPA) and attendance. Severity: NA.	Level of study; enrolment type; if students have dependents.	Type: NR. Effectiveness: NR.	Time management challenges, lack of awareness of available resources, and social stigma associated with seeking help for family or work-related challenges.	A variety of academic, non-academic, and personal factors such as the challenges faced by students should be considered concerning factors that may affect attendance and student performance in higher education.
Kent et. al. (2021) [23]	Population: A non-random sample of University of Tasmania students (n = 1858); racial breakdown not provided Study design/approach: Cross-sectional, online survey/quantitative	To characterize university students’ perceptions of the importance of sustainable foods. To determine the relationship between perceptions and the frequency of purchasing sustainable foods.	Type: Food insecurity. Severity: Mild to moderate (38%).	Female and older students were more likely to perceive sustainable food as important and purchase them frequently.	Type: Some university sustainability programs existed but were underutilized or lacked coverage across all campuses. Effectiveness: On-campus initiatives promoting local and sustainable food were found to have a moderate influence.	The cost of sustainable food, the availability of sustainable options on campus, and limited awareness about sustainability programs were major barriers.	Students who purchase sustainable foods frequently are more likely to be female, older, and food insecure.
Moscaritolo, L. B., et al. (2022) [33]	Oceania subset Population: Enrolled university students in the Oceania region (n = 108); racial breakdown not provided Study design/approach: Cross-sectional/quantitative Overall study 781 Student Affairs and Services (SAS) practitioners from around the world; exploratory survey study	To report on the Student Affairs and Services (SAS) mitigation strategies to reduce the impact of COVID-19 on students, in general, and international students specifically.	Oceania subset Type: Mental well-being; and financial hardships; discrimination. Severity: High emotional stress (96%); high financial challenges (74%); international students experiencing discrimination (25%). Overall study International students were the most impacted group, facing challenges such as emotional stress (96%), inability to return home (88%), financial hardship (74%), and fear (67%).	Oceania subset International student status, especially Asian and Chinese students; low SES; and marginalized groups. Overall study Border closures; loss of part-time jobs; financial difficulties; and emotional strain due to isolation and discrimination (particularly Asian students).	Oceania subset Type: A dedicated cross-functional team working with international students and communicating with them via phone, group chat, workshops, and Moodle (Oceania); emergency grants to students for transportation, housing, and food. Effectiveness: NR. Overall study Emergency grants, mental health services, counselling, and accommodations were critical. Communication and peer support systems were effective in some regions.	Limited access to government support, lack of information about available services, language barriers, and stigma around seeking help.	The findings from this study have implications for guiding higher education and SAS decision-makers to improve the support SAS provides to international students. The long-term restructuring of SAS is suggested to enhance international student support post pandemic.
Murray, S., et al. (2021) [26]	Population: Enrolled University of Tasmania students (n = 1858); racial breakdown not provided Study design/approach: Cross-sectional online survey/quantitative	To determine the prevalence of food insecurity and its relationship with the satisfaction of on-campus food choices.	Type: Food insecurity. Severity: Moderate (38.1%). High dissatisfaction (47%) was noted among food-insecure students regarding affordable food options on campus.	Enrolment status (international students); younger age; cost of food.	Type: Range of food availability and affordability on campus. Effectiveness: Moderate (33–37%).	Poor awareness of available food support services.	Food insecurity and deficits in the university food environment are highly prevalent. This can inform the development of strategies to improve the food available on campus, including affordable, sustainable, and local options.
Ballingall et. al. (2016) [18]	Population: First-year university students (aged 18–21) enrolled in a metropolitan Melbourne university (n = 9). (recently moved out of family home); racial breakdown not provided Study design/approach: Interviews/qualitative study	To explore the factors that affect young adults’ environment such as family background, education and skills, living conditions, and social environment.	Type: Healthy eating and nutrition; Financial challenges. Severity: NA.	Lack of prior exposure to food shopping, meal preparation, and budgeting skills; balancing studies, work, and social commitments affected food choices.	Type: Some informal peer support for meal preparation and food sharing existed, but structured support services were not identified. Effectiveness: NA.	NR	This study suggests further research on the role of social support in the Australian context and offers tertiary education providers initial considerations for evaluating if their on-campus accommodations promote healthy eating habits.
Shi, Y. and M. Allman-Farinelli (2023) [28]	Population: Students attending a Sydney university (n = 467; 376 domestic and 91 international); 1.3% Indigenous Australians Study design/approach: Cross-sectional survey/quantitative	To explore the prevalence, sociodemographic determinants, and effects of food insecurity among domestic and international students during the COVID-19 pandemic.	Type: Food insecurity. Severity: Low–moderate food insecurity; 13% of domestic students and 18.7% of international students were food-insecure.	International student status; undergraduate student status; living away from home/changes in living arrangements due to the pandemic; loss of employment due to COVID-19.	Type: Food assistance. Effectiveness: NR.	Stigma, limited access to government support for international students, and inadequate awareness of food assistance services.	International students are at a higher risk of food insecurity than domestic students. Mandatory policies to improve campus food environments, increased financial support for food access, and the better dissemination of food-related knowledge including sourcing foods, food relief programmes, nutrition, cooking skills, and recipes are needed to address these inequities.
Bexely, E., et al. (2013) [9]	Population: Enrolled students from 37 Universities in Australia (n = 11,761); 7.2% Indigenous Australians Study design/approach: Cross-sectional online survey/quantitative	To provide an evidence-based understanding of the financial circumstances of the student population in Australia (international and domestic) concerning access to income support and scholarships, income from paid employment, and the impact of paid work on study, study and living costs, and student debt.	Type: Financial difficulty. Severity: High financial difficulty (68% of full-time students and 64% of part-time students).	Indigenous status; international students; low SES; combining work and study.	Type: Student income support from the Australian government; university student services. Effectiveness: NR.	Austudy application rejection (21%) and a belief that Austudy (government financial assistance program) application would be unsuccessful.	Recommendations include increasing financial aid for low-income and Indigenous students, improving work–study balance, and revisiting policies for income support to ensure they address the growing diversity of the student population.
Corney et. al. (2024) [32]	Population: Six state-wide (Victoria) international student associations (n = 375); 27% from China, 16% from India, 13% from Nepal, and 27% from other Asian countries Study design/approach: Mixed methods/qualitative and quantitative	To explore accommodation, subjective well-being, mental health help-seeking, and strategies for promoting well-being among international students during COVID-19.	Type: Personal Well-Being Index (PWI); accommodation (qualitative). Severity: Low personal well-being index in 59% of students.	Safety of accommodation; cost of accommodation; exposure to a new culture in Australia; distance from familiar family and friend networks and the impact of COVID-19 lockdowns.	Type: Regularly helping others (volunteering) (contributed 2.3% PWI variance); physical activity (1.2% PWI variance); social connection (1.1% PWI variance). Effectiveness: All well-being measures were strong predictors of high PWI.	Cost of accessing support, language and cultural barriers, lack of information on where to find support; and stigma.	The study’s findings have implications on informing policy and practice in service and facility provision regarding well-being, connectedness, and help-seeking for mental health support.
Baglow and Gair (2019) [30]	Population: 2320 students from the 29 accredited social work programs in Australia; 4% Indigenous Australians Study design/approach: Descriptive/qualitative and quantitative	To explore the impact of low levels of income on the lives and study success of a cohort of Australian social work students.	Type: Financial stress. Severity: High financial stress (55–73%).	Rising living costs; long hours in paid employment; and low levels of income support, especially during compulsory field placements.	Type: NR. Effectiveness: NR.	NR	There is a need for advocacy for increased support for tertiary social work students.
Mihrshahi, S., et al. (2022) [25]	Population: Students enrolled at Macquarie University in Sydney, Australia (n = 105 (66 domestic and 39 international)); racial breakdown not provided Study design/approach: Cross-sectional/quantitative	To explore the prevalence of food insecurity and psychological distress during the COVID-19 pandemic, with a focus on international students.	Type: Food insecurity status; mental health (psychological distress); sleep. Severity: High level of food insecurity (42%); high psychological distress (52.2%, with high and very high levels); poor sleep quality was reported in 24% of the study population.	The impact of COVID -19 lockdowns; financial hardship due to job loss; and lack of government support for international students.	Type: Some universities provided emergency financial assistance and food vouchers, but international students had limited access to sustained financial support. Effectiveness: NR.	Ineligibility for government financial aid, limited awareness of available support services, and the stigma associated with using food relief.	The study’s findings may help governments and educational institutions design appropriate support, particularly financial and psychological, for both international and domestic university students.
Micevski, D. A., et al. (2014) [24]	Population: University students enrolled at Deakin University, Victoria, Australia (n = 124); racial breakdown not provided Study design/approach: Cross-sectional/quantitative	To assess the prevalence, severity, and potential determinants of food insecurity among tertiary students.	Type: Food insecurity. Severity: Mild–moderate food insecurity (18–30%).	Receiving government support; living away from family.	Type: NR. Effectiveness: NR.	Lack of knowledge of university support services.	Food insecurity without hunger is a significant problem for students at Deakin University. There is a need to increase availability and accessibility at Deakin University, with one possibility being the establishment of on-campus food banks.
Kent. et. al. (2024) [22]	Population: Enrolled students at the University of Wollongong (n = 197); racial breakdown not provided Study design/approach: Cross-sectional, online survey/quantitative	To identify groups of Australian university students at an increased risk of food insecurity at the University of Wollongong and their engagement with on-campus food initiatives. To determine the relationship between food insecurity and a validated index of diet quality.	Type: Food security. Severity: Moderate to high food insecurity (54%).	Living situation (on campus/renting or shared); being male; increased cost of living and food prices.	Type: A food pantry (open for 1.5 h twice a week during the semester); and a community garden. Effectiveness: NR.	Stigma, lack of awareness of services, limited engagement with campus food initiatives (like the community garden), and difficulties in accessing nutritious food due to high costs.	Food-insecure students had notably poorer diet quality, especially in their intake of fruits and vegetables. This study highlights the pressing need for universities to develop comprehensive food policies and strategies. These should focus on improving the on-campus food environment to tackle the root causes of food insecurity and poor diet quality, ensuring all students have equitable access to healthy food.
Hughes, R., et al. (2011) [20]	Population: Students from a Queensland university (n = 399); racial breakdown not provided Study design/approach: Cross-sectional/quantitative	To describe the prevalence, distribution, and severity of food insecurity, and related behavioural adaptations, among a sample of Australian university students.	Type: Food Insecurity. Severity: Moderate–severe food insecurity (13–47%).	Shared accommodation; low income and receiving government support; financial stress; working long hours alongside studies.	Type: Coping strategies from students. Effectiveness: Living with parents; working more than 10 h per week outside of university; and borrowing money and food.	Stigma, lack of awareness about food relief services, and difficulty balancing work and study contributed to the underutilization of support.	University students are at significant risk of food insecurity, which is attributed, in part, to inadequate income support.
Bennett, C. J., et al. (2022) [31]	Population: Undergraduate and postgraduate university students enrolled with an Australian university, Monash University, at its Australian (multi-site) or Malaysian campuses (n= 1315); racial breakdown not provided Study design/approach: Cross-sectional/quantitative	To examine the prevalence, severity, coping strategies, and precipitating factors of food insecurity during the COVID-19 pandemic.	Type: Food insecurity associated with deteriorating mental health. Severity: Moderate (32%).	International student status; unemployment; looking for work; living alone; and enrolment in a postgraduate degree.	Type: Emergency food relief, financial assistance from organizations, and subsidized meals were accessed. Effectiveness: These services were not always sufficient.	COVID-19 lockdown restrictions limited access to family/friends for food, physical access to stores, and availability of culturally appropriate food.	The study provides recommendations for increasing financial support for students, improving access to food (including culturally appropriate options), and the better integration of mental health services to address the psychological impacts of food insecurity.

^1^ Financial difficulty levels were categorized by prevalence: “Low” (<25%), “Moderate” (25–50%), and “High” (>50%), indicating increasing severity in challenges meeting basic needs, from minimal disruptions to severe strain requiring emergency aid.

## 4. Discussion

This scoping review highlights the extensive financial challenges facing Australian and New Zealand university students, affecting their well-being and academic performance, particularly among international students. These international students encounter additional challenges, including restricted access to support services, limited job opportunities, and cultural or language barriers. The findings from the literature suggest financial difficulties are widespread among students, and the impact on well-being is consistently negative, with many students experiencing food and housing insecurity as well as psychological distress [21,23]. However, evidence on effective strategies to alleviate these hardships is limited.

### 4.1. Key Financial Hardships

Food and housing security were the primary financial hardships experienced by students. This was more evident for international students; however, it was still a concern for domestic students. The situation appears to have been exacerbated by recent global events including the global financial crisis from 2006 to 2008 and the COVID-19 pandemic, making it harder for students to afford nutritious food and decent housing. Additionally, changes at the government and policy levels have affected the cost of studying in terms of increases and structural changes to fees [36]. These have become increasingly complex, adding to students’ stress relating to financial challenges.

### 4.2. Current Campus Support and Effectiveness

Australian universities provide various support services, including financial aid, food assistance, mental health services, and emergency grants, but evidence of their effectiveness remains limited [19,32]. A recurring issue highlighted across studies is that these support mechanisms often fail to meet demand or fully alleviate students’ financial pressures [21,33]. Additionally, services are often underutilized due to a lack of awareness, stigma, and accessibility barriers, particularly for international students, who may face additional challenges due to restrictive eligibility criteria or cultural differences [26,28]. Although universities offer essential services, further scrutiny and refinement are needed to improve the effectiveness and accessibility of these interventions.

### 4.3. Barriers to Access

Many students encounter difficulties accessing available support services due to complex or fragmented service delivery. Studies suggest that students often struggle to locate or navigate university support systems, underscoring the need for streamlined, accessible resources and clear communication [20,24]. Additionally, the stigma associated with seeking assistance, logistical constraints (e.g., limited operating hours), and cultural or linguistic barriers further inhibit service utilization, especially for international students [31,32]. To address these issues, universities must develop inclusive, user-friendly support pathways that effectively serve all students in need.

### 4.4. Policy Dependence and Systemic Limitations

Support for financially distressed students remains highly dependent on federal policy, limiting universities’ ability to implement substantial, long-term solutions [10,35]. Changes in government priorities and funding allocations create an unstable support environment, leaving students vulnerable to policy shifts. The commonly cited “I did it tough” attitude in public discourse can obscure the fact that current students face unique, compounded financial challenges, such as rising costs of living, and the prolonged effect of the global pandemic that previous generations did not encounter. Given the implications of financial stress on academic completion rates and broader workforce development [37], a community-wide conversation on sustainable student support is both timely and necessary.

Addressing student financial challenges requires multilayered interventions across individual, interpersonal, organizational, community, and policy levels as demonstrated by the socioecological model [38]:Individual Level: Enhancing Financial Literacy and Resilience

At the individual level, universities play an integral role in equipping students with essential life skills to manage financial stress. There is a need for targeted programs focusing on financial literacy, mental health, and nutrition knowledge that can foster resilience and improve decision making. Educating students about budgeting, debt management, and financial planning helps alleviate the stress associated with limited financial resources [26]. Workshops, online modules, and one-on-one financial counselling can provide practical and accessible resources.

Additionally, financial stress often correlates with poor mental health and dietary inadequacies. Initiatives to enhance mental health awareness and nutrition knowledge can empower students to maintain well-being despite financial constraints [19]. Such interventions must be tailored to diverse student needs, ensuring accessibility for international, first-generation, those living with a disability, and socioeconomically disadvantaged students.

Interpersonal Level: Fostering Social Support Networks

On an interpersonal level, there is an emphasis on the importance of social connections in mitigating financial stress. Peer support groups and mentorship programs can provide both emotional and practical support. Student-led support groups offer shared understanding and informal advice, which can be particularly valuable for international students facing cultural and social isolation [21]. Organizing and providing mentorship programs that connect students with alumni or senior peers can also provide guidance on navigating financial challenges, accessing resources, and balancing academic demands [20]. These interpersonal networks strengthen community ties and build a sense of belonging, which is crucial for students’ mental health and academic persistence.

Organizational Level: Building a Supportive Campus Infrastructure

At the organizational level, evidence suggests that universities need to prioritize proactive and inclusive measures to address financial hardship. Including students’ perspectives in designing support services ensures that interventions remain relevant and effective [21]. Universities should conduct regular assessments of students’ well-being and, where necessary, expand scholarships, emergency grants, and affordable on-campus housing, particularly for vulnerable groups such as students from low socioeconomic backgrounds. Additionally, mental health services and food support initiatives, such as subsidized meal plans or campus food banks, are essential to create a safety net for students [32].

Beyond financial support, universities must consider the structure of teaching and learning to provide flexibility for students who are managing work and study commitments. Initiatives such as offering hybrid or asynchronous course options [39] consider scheduling some classes during evenings or weekends, and allowing for part-time study pathways can help students balance employment responsibilities without compromising their academic success [40]. These approaches not only support students’ financial well-being but also foster greater accessibility and inclusivity within the academic environment.

By incorporating flexible learning options alongside financial and mental health support, universities can create an equitable academic infrastructure. Involving students in program design and fostering advocacy skills further empowers them, increasing agency and ensuring solutions remain relevant and impactful. These efforts collectively reflect a commitment to fostering a supportive academic environment that addresses both immediate financial challenges and long-term student success.

Community Level: Collaborating with External Partners

Addressing financial hardship extends beyond campus boundaries, requiring collaboration with community partners. Partnerships with local governments, non-profits, and housing organizations can increase the availability of affordable housing options for students. This is particularly important given the rising costs of rent in urban centres. In addition, collaborations with food banks, health clinics, and legal aid organizations can supplement campus resources and provide comprehensive support [28]. These community-level interventions expand the support network for students, addressing external stressors that impact academic success.

Policy Level: Advocating for Systemic Change

At the policy level, universities must advocate for systemic reforms to stabilize and enhance student support structures. Federal funding packages, social protections for international students, and affordable healthcare access can significantly alleviate financial burdens [41]. Routine national surveys on student well-being provide critical insights into emerging challenges and inform evidence-based policies [10]. Through sustained advocacy and engagement with policymakers, universities can drive long-term improvements in the higher education landscape.

## 5. Conclusions

This review offers a comprehensive synthesis of the current evidence on financial challenges among students in Australia and New Zealand. However, there are limitations worthy of note. The exclusion of non-English language studies and the grey literature may have omitted relevant insights on support services. Additionally, the reliance on cross-sectional data restricts our understanding of the long-term efficacy of interventions. There is also some variability in study populations, with some studies focusing solely on undergraduates, while others include postgraduates and international students, making comparisons challenging. Another key limitation is the inconsistency in the measurement tools used across studies, leading to variations in how financial hardship and food insecurity are defined and assessed. Furthermore, this review does not extensively examine institutional and policy-level factors, missing insights into how government policies and university funding impact student financial well-being.

To build on these findings, future research should adopt longitudinal approaches to track the long-term effectiveness of support services and explore how financial challenges evolve over time. It is also essential to examine students’ perspectives on academic and personal challenges, particularly in a post-pandemic context, and assess the impact of academic workload on financial stress and student support needs. Additionally, incorporating qualitative research would provide a deeper understanding of students’ lived experiences, ensuring that support mechanisms are tailored to their diverse needs.

## Figures and Tables

**Figure 1 ijerph-22-00356-f001:**
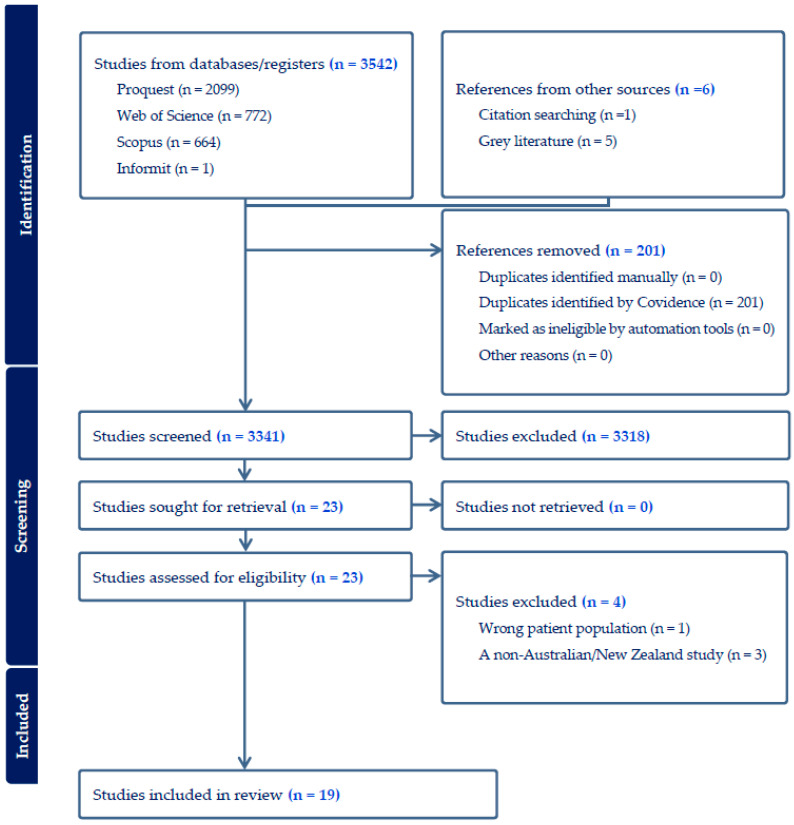
PRISMA flow diagram of record identification and study selection.

## Data Availability

The original contributions presented in this study are included in the article. Further inquiries can be directed to the corresponding author.

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
