# Peer review of "Exploring Financial Challenges and University Support Systems for Student Financial Well-Being: A Scoping Review"

_ijerph, 2025, doi:10.3390/ijerph22030356_

Round 1

Reviewer 1 Report

Comments and Suggestions for Authors

This was an interesting paper and, as appropriate for a review of previously published work, provided an excellent summary of the financial challenges facing higher education students in Australia and New Zealand based on the included papers.  It also summarized the various impacts of financial stress such as financial uncertainty, stress, food insecurity, mental and physical health problems, etc.   This material clearly aligns with similar work from other countries. 

Beyond this strong summary relevant to Australia and New Zealand, the authors take the issue to the natural next step in identifying the various efforts taken by Institutions of Higher Education (IHEs) to address these challenges.  This is the real value add, in my opinion.  The authors highlight findings related to the what IHEs are doing (or not) in response through support services, and the impact of those efforts.  In particular, I appreciated the finding that more often than not, these IHE support services appear to be ineffective.  The example of the limitations of on-campus food pantries, often run by students themselves, or community gardens, which can provide a valuable teaching opportunity but usually are not a replacement for stable financial resources, rang clear and true to me from several decades of research.  This raises important questions on the role of IHEs vis-a-via government and society in providing support for students.  

The authors have brought us up to speed on the current research on student financial stress from this area of the world.  They have then taken the next step to clearly challenge IHEs in particular and society in general to improve how we respond. 

I have two minor comments -First,  I recommend more detail be provided on the reasons why the vast majority of potential studies were cut down. There seems to be a reference to a table that will so this on lines 233-235, but I did not see that table - my apologies if I am reading the reference incorrectly.  Regardless, I was somewhat confused on how the authors went from well over 3000 articles to several dozen.  It seems unnecessary that you would start with such a large number.  Second, there is a typo on line 243.  

Author Response

Comments 1: This was an interesting paper and, as appropriate for a review of previously published work, provided an excellent summary of the financial challenges facing higher education students in Australia and New Zealand based on the included papers.  It also summarized the various impacts of financial stress such as financial uncertainty, stress, food insecurity, mental and physical health problems, etc.   This material clearly aligns with similar work from other countries. 

Response 1: Thank you for pointing this out. We agree with this comment.

Comments 2: Beyond this strong summary relevant to Australia and New Zealand, the authors take the issue to the natural next step in identifying the various efforts taken by Institutions of Higher Education (IHEs) to address these challenges.  This is the real value add, in my opinion.  The authors highlight findings related to the what IHEs are doing (or not) in response through support services, and the impact of those efforts.  In particular, I appreciated the finding that more often than not, these IHE support services appear to be ineffective.  The example of the limitations of on-campus food pantries, often run by students themselves, or community gardens, which can provide a valuable teaching opportunity but usually are not a replacement for stable financial resources, rang clear and true to me from several decades of research.  This raises important questions on the role of IHEs vis-a-via government and society in providing support for students.  

Response 2: Thank you for pointing this out. We agree with this comment.

Comments 3: The authors have brought us up to speed on the current research on student financial stress from this area of the world.  They have then taken the next step to clearly challenge IHEs in particular and society in general to improve how we respond. 

Response 3: Thank you for pointing this out. We agree with this comment.

Comments 4: I have two minor comments -First,  I recommend more detail be provided on the reasons why the vast majority of potential studies were cut down. There seems to be a reference to a table that will so this on lines 233-235, but I did not see that table - my apologies if I am reading the reference incorrectly.  Regardless, I was somewhat confused on how the authors went from well over 3000 articles to several dozen.  It seems unnecessary that you would start with such a large number. 

Response 4: Thank you for pointing this out. We agree with this comment.

Therefore, we have updated the manuscript to include a supplementary table on reasons for excluding studies.

Comments 5: Second, there is a typo on line 243.  

Response 5: Thank you for pointing this out. We agree with this comment.

Therefore, we have corrected the typo on line 243 (page 7) to read “studies” instead of “study.”

Reviewer 2 Report

Comments and Suggestions for Authors

This was a well written article. The metholody used is strong. My main suggestion is in the introduction, I think the focus should be on the synthesis of the available services and that is the gap the scoping review is aiming to address. See below for some more detailed suggestions: 

Line 83-88: Please add citations, particularly for the definition of food insecurity. For the definition, I would change "buy" to "access and availability" as that is more accurate. 

Line 92-100: Generally, not very clear on what the gap is that the scoping review is aiming to address - the aim needs to be stregthened. I would emphasize the fact that this paper aims to synthesize the services provided since that is the unique piece compared to the finanicial impacts, because that is well documented in the literature. 

Line 133-134: Include explanation why food insecurity and housing affordability/avilability were included. Was it because they were considered related factors based on literature? 

Line 145-147: I suggest removing since it is the opposite population of what was listed in the inclusion criteria. 

Line 302: Include how many studies implemented services and how many tested the effectiveness of these services. 

Table 1: Include the racial breakdown under population and study design. 

Discussion: Were there any limitations in the study design/analysis of the manuscripts? 

Author Response

This was a well written article. The metholody used is strong. My main suggestion is in the introduction, I think the focus should be on the synthesis of the available services and that is the gap the scoping review is aiming to address. See below for some more detailed suggestions: 

Comments 1: Line 83-88: Please add citations, particularly for the definition of food insecurity. For the definition, I would change "buy" to "access and availability" as that is more accurate. 

Response 1: Thank you for pointing this out. We have included a citation on line 88 and changed “buy” to “access and availability” on lines 83-84- highlighted in red.

Comments 2: Line 92-100: Generally, not very clear on what the gap is that the scoping review is aiming to address - the aim needs to be stregthened. I would emphasize the fact that this paper aims to synthesize the services provided since that is the unique piece compared to the finanicial impacts, because that is well documented in the literature. 

Response 2: Thank you for pointing this out. We have edited lines 92-100 (highlighted in red, to emphasize that given the well-documented financial challenges faced by university students, this scoping review goes beyond summarizing financial impacts by focusing on synthesizing the services and support strategies implemented by universities in Australia and New Zealand to address these challenges.

Comments 3: Line 133-134: Include explanation why food insecurity and housing affordability/avilability were included. Was it because they were considered related factors based on literature? 

Response 3: Thank you for pointing this out. We have included an explanation with citation on why food insecurity and housing affordability/availability were included – highlighted in red. This is because they are widely recognised in the literature as key indicators of financial hardship among university students.

Comments 4: Line 145-147: I suggest removing since it is the opposite population of what was listed in the inclusion criteria. 

Response 4: Thank you for pointing this out. We deleted the exclusion criterion on population on lines 145-147.

Comments 5: Line 302: Include how many studies implemented services and how many tested the effectiveness of these services. 

Response 5: Thank you for pointing this out. We have included some more information on lines 300-304, highlighted in red, on the number of studies that explored support services and the subsequent effectiveness where reported.

Comments 6: Table 1: Include the racial breakdown under population and study design. 

Response 6: Thank you for pointing this out. In most Australian studies, racial breakdowns of the sample population are not typically reported, though Indigenous status is often included. To ensure completeness, we have incorporated all available data on both Indigenous status and racial demographics in Table 1.

Comments 7: Discussion: Were there any limitations in the study design/analysis of the manuscripts? 

Response 7: Thank you for pointing this out. We have edited the conclusion to include some more limitations (lines 480-487).

Reviewer 3 Report

Comments and Suggestions for Authors

Reviewer Assessment of the entire Manuscript

 University students' financial challenges have emerged as a pressing concern amid fluctuating global economic conditions. Rising tuition fees, escalating living costs, and economic instability have placed immense pressure on students, adversely affecting their academic performance and overall well-being. Vulnerable groups, such as students from low socioeconomic backgrounds and international students, are disproportionately impacted due to insufficient financial support and limited access to employment opportunities.

Universities have attempted to mitigate these challenges through initiatives like emergency grants, food pantries, and community gardens. However, the effectiveness of these measures remains limited, with many students continuing to face food and housing insecurity alongside psychological distress. The COVID-19 pandemic has further exacerbated these financial difficulties, underscoring the urgent need for comprehensive strategies to enhance student financial well-being and address systemic issues within government and educational policies.

In conclusion, university students' financial challenges, intensified by rising costs and economic instability, significantly hinder their academic performance and well-being. Addressing these issues demands a holistic approach, combining institutional support, policy reforms, and systemic changes to ensure equitable access to education and long-term financial stability for all students.

Title: Exploring financial challenges and university support for student financial well-being: A scoping review

Your proposed title is straightforward and comprehensive, effectively summarizing the study's focus. It highlights students' financial challenges and the support measures universities provide, clarifying what the scoping review aims to address. This title should capture the interest of readers keen to understand students' financial well-being and the role of university support systems.

How about this revision of the title:

"Navigating Financial Challenges and University Support Systems for Student Well-Being: A Scoping Review"

This title maintains the clarity and focus of the original while emphasizing the dual aspects of financial challenges and support systems.

Abstract

Here are three suggestions that could help improve the abstract of this study:

  • Clarify the Scope of "Vulnerable Groups": Specify the vulnerable groups mentioned in the results and conclusion sections to understand better which student populations are most affected.
  • Elaborate on Methods: Briefly explain why the Arskey and O'Malley framework was chosen for this scoping review to give context to the methodological approach.
  • Highlight Key Findings with Specific Data: Instead of stating "over half (8% - 68%) of students face significant financial issues," specify a few key statistics from the included studies to strengthen the results section and emphasize the review's findings.

Keywords

Reduce the keywords; they are too many. The keywords can be five in number.

Introduction

The introduction section requires a more substantial build-up. It should highlight how the financial landscape for university students has become increasingly complex, with rising tuition fees and living costs compounding the pressures of economic instability. These financial challenges hinder academic performance and profoundly impact students' overall well-being, underscoring the urgent need for universities to implement effective support systems. Below are five questions to help strengthen the introduction:

  • What financial challenges do university students encounter due to global economic shifts?
  • How have increasing tuition fees and living costs affected students' academic performance and well-being?
  • What were the effects of the 2007-2009 global financial crisis on university students, and how do these compare to the impacts observed during the COVID-19 pandemic?
  • How do financial challenges vary among students from different socioeconomic backgrounds and geographic locations?
  • What measures have universities in Australia and New Zealand introduced to assist students facing financial challenges, and how effective have these initiatives been?

Discussion

Remove all headings or themes within the discussion section to allow the narrative to flow seamlessly. Below are six guiding questions to enhance the discussion section. These questions aim to deepen the analysis, address gaps, and provide a more comprehensive understanding of the topic.

  • What are the root causes of the financial challenges experienced by university students, and how do these challenges differ between domestic and international students?
  • How do financial challenges affect student life, including academic performance, mental health, and social relationships?
  • What specific strategies have universities introduced to assist financially struggling students, and what has been the effectiveness of these measures?
  • How do the financial challenges faced by students from low socioeconomic backgrounds compare to those experienced by international students?
  • What are the limitations in the existing literature on the financial challenges university students face, and how can future research address these gaps?
  • What policy recommendations can be derived from the findings of this scoping review to support financially vulnerable students more effectively?

Strengths and Limitations

What are the strengths and limitations of this study? Please ensure they are clearly outlined and incorporated into the manuscript. Kindly revise.

 What are the implications of the study findings?

What are the implications of the study findings? In research, implications refer to the conclusions that can be drawn from the study's results, which may have either theoretical significance or practical applications. Kindly revise.

Contributions of this study

What unique contributions does this study make to the existing body of research, particularly compared to studies conducted in developed and developing nations?

Future suggestions for prospective studies

What suggestions does your study offer for future research, particularly for replication in different settings? Future research directions are insights from the study's findings that highlight potential areas for further exploration. These can include addressing the current study's limitations or exploring gaps that emerged during the research process.

Need a Professional English Editor for Editing and Proofreading

The manuscript requires the expertise of a Professional English Editor to edit and proofread the text thoroughly. This will ensure all grammatical errors are corrected, and the lexical structure is polished for clarity and coherence.

References

Do the references in this study adhere to the journal's required format? Please review and revise them as necessary to ensure full compliance.

Article Suggestions: [This can help you model your paper]

Akokuwebe ME, Palamuleni ME and Idemudia ES (2025) Population dynamics and digitalization: implications for COVID-19 data sources in South Africa—a scoping review. Front. Public Health 12:1537057. doi: 10.3389/fpubh.2024.1537057.

Comments on the Quality of English Language

Need a Professional English Editor for Editing and Proofreading

The manuscript requires the expertise of a Professional English Editor to edit and proofread the text thoroughly. This will ensure all grammatical errors are corrected, and the lexical structure is polished for clarity and coherence.

Author Response

University students' financial challenges have emerged as a pressing concern amid fluctuating global economic conditions. Rising tuition fees, escalating living costs, and economic instability have placed immense pressure on students, adversely affecting their academic performance and overall well-being. Vulnerable groups, such as students from low socioeconomic backgrounds and international students, are disproportionately impacted due to insufficient financial support and limited access to employment opportunities.

Universities have attempted to mitigate these challenges through initiatives like emergency grants, food pantries, and community gardens. However, the effectiveness of these measures remains limited, with many students continuing to face food and housing insecurity alongside psychological distress. The COVID-19 pandemic has further exacerbated these financial difficulties, underscoring the urgent need for comprehensive strategies to enhance student financial well-being and address systemic issues within government and educational policies.

In conclusion, university students' financial challenges, intensified by rising costs and economic instability, significantly hinder their academic performance and well-being. Addressing these issues demands a holistic approach, combining institutional support, policy reforms, and systemic changes to ensure equitable access to education and long-term financial stability for all students.

Title: Exploring financial challenges and university support for student financial well-being: A scoping review

Your proposed title is straightforward and comprehensive, effectively summarizing the study's focus. It highlights students' financial challenges and the support measures universities provide, clarifying what the scoping review aims to address. This title should capture the interest of readers keen to understand students' financial well-being and the role of university support systems.

Comments 1: How about this revision of the title:

"Navigating Financial Challenges and University Support Systems for Student Well-Being: A Scoping Review"

This title maintains the clarity and focus of the original while emphasizing the dual aspects of financial challenges and support systems.

 Response 1: Thank you for your suggestion. We have edited the title of our manuscript to “Exploring Financial Challenges and University Support Systems for Student Financial Well-being: A Scoping Review.” This title keeps clarity, precision, and a structured focus while incorporating the strengths of both titles.

Abstract

Here are three suggestions that could help improve the abstract of this study:

  • Comments 2: Clarify the Scope of "Vulnerable Groups": Specify the vulnerable groups mentioned in the results and conclusion sections to understand better which student populations are most affected.

Response 2: Thank you for this comment. Lines 41-42 specify the vulnerable groups mentioned.

  • Comments 3: Elaborate on Methods: Briefly explain why the Arskey and O'Malley framework was chosen for this scoping review to give context to the methodological approach.

Response 3: Thank you for this comment. The abstract is limited due to word count. The reason for using the Arskey and O'Malley framework is explained within the methods sections of the manuscript and briefly in the abstract (Lines 33-34).

  • Comments 4: Highlight Key Findings with Specific Data: Instead of stating "over half (8% - 68%) of students face significant financial issues," specify a few key statistics from the included studies to strengthen the results section and emphasize the review's findings.

Response 4: Thank you for this comment. We have added extra statistics to the result section to highlight key findings.

Keywords

Comments 5: Reduce the keywords; they are too many. The keywords can be five in number.

Response 5: Thank you for this comment; we have included the top five keywords.

Introduction

The introduction section requires a more substantial build-up. It should highlight how the financial landscape for university students has become increasingly complex, with rising tuition fees and living costs compounding the pressures of economic instability. These financial challenges hinder academic performance and profoundly impact students' overall well-being, underscoring the urgent need for universities to implement effective support systems. Below are five questions to help strengthen the introduction:

  • Comments 6: What financial challenges do university students encounter due to global economic shifts?

Response 6: Thank you for your suggestion. Lines 55-63 outline the financial challenges students face due to global economic shifts.

  • Comments 7: How have increasing tuition fees and living costs affected students' academic performance and well-being?

Response 7: Thank you for your suggestion. Lines 61-67 describe evidence from literature, demonstrating how financial challenges affected students’ predisposition towards pursuing higher qualifications. From an Australian perspective, data shows that 13% of students were forced to go without necessities.  

  • Comments 8: What were the effects of the 2007-2009 global financial crisis on university students, and how do these compare to the impacts observed during the COVID-19 pandemic?

Response 8: Thank you for your suggestion. Lines 62-80 provide evidence from the literature of how the 2007–2009 Global Financial Crisis (GFC) affected students' welfare, highlighting financial struggles such as limited job opportunities, increased financial stress, and reduced access to necessities. The introduction further explains how these challenges were exacerbated by the COVID-19 pandemic, which led to widespread job losses, heightened food insecurity, and severe mental health impacts, particularly for international and low-SES students. The comparison underscores how COVID-19 had a more immediate and widespread effect, intensifying financial and social hardships beyond those experienced during the GFC.

  • Comments 9: How do financial challenges vary among students from different socioeconomic backgrounds and geographic locations?

Response 9: Thank you for this comment. Lines 67-74 describe how the impacts of financial challenges outlined earlier in the manuscript were more evident for students from regional locations and students from low SES backgrounds doubly affected. Lines 71-73 read “This has been attributed to a complex interplay of factors, including the costs of study materials, travel to university, usual expenses of living, and sometimes supporting a family, which contribute to the financial burden and challenges experienced”

  • Comments 10: What measures have universities in Australia and New Zealand introduced to assist students facing financial challenges, and how effective have these initiatives been?

Response 10: Thank you for this comment. An aim of this study was to examine the measures introduced by universities to address students’ financial challenges. As such, this question cannot be fully answered in the introduction, as the effectiveness of these initiatives is evaluated throughout the study based on the evidence from the scoping review.

Discussion

Comments 11: Remove all headings or themes within the discussion section to allow the narrative to flow seamlessly.

Response 11: Thank you for your suggestion. However, the headings within the discussion section are essential as they structure the analysis using the socioecological model, which is fundamental to our approach. This model allows us to examine financial challenges at multiple levels—individual, interpersonal, organisational, community, and policy—ensuring a comprehensive and systematic exploration of the issue. Removing the headings would disrupt the logical flow and clarity of the discussion, making it more difficult for readers to follow how financial hardship impacts students across different dimensions. The structured approach also enhances policy and intervention recommendations, aligning with the study's objectives.

Below are six guiding questions to enhance the discussion section. These questions aim to deepen the analysis, address gaps, and provide a more comprehensive understanding of the topic.

  • Comments 12: What are the root causes of the financial challenges experienced by university students, and how do these challenges differ between domestic and international students?

Response 12: Thank you for your comment. Lines 347-349; and 355-363 identify the key financial hardships and highlight rising living costs, limited employment opportunities, and inadequate financial aid as key causes of student financial hardship. Domestic students have some government support, while international students face higher tuition, visa work restrictions, and lack of aid, worsening their struggles.

  • Comments 13: How do financial challenges affect student life, including academic performance, mental health, and social relationships?

Response 13: Thank you for pointing this out. The discussion section of the manuscript highlights that financial challenges significantly impact student life, affecting academic performance, mental health, and social relationships. Many students struggle with balancing work and study, leading to missed classes, poor grades, and limited access to study resources. Financial stress contributes to anxiety, depression, and food insecurity, with international and low-SES students being the most vulnerable. Housing instability and social isolation further deteriorate well-being, as students withdraw from social activities due to financial constraints. The study emphasizes the need for stronger university support systems, including financial aid, mental health services, and policies addressing food and housing insecurity.

  • Comments 14: What specific strategies have universities introduced to assist financially struggling students, and what has been the effectiveness of these measures?

Response 14: Thank you for pointing this out. Lines 366-374 in our manuscript outlines current campus support and effectiveness.

  • Comments 15: How do the financial challenges faced by students from low socioeconomic backgrounds compare to those experienced by international students?

Response 15: Thank you for highlighting this point. It is indeed an important research question that warrants in-depth qualitative exploration. This is one of the recommendations outlined in our study (lines 486-489). Most of the articles included in our scoping review primarily utilise quantitative methodologies, which highlights that students from low-SES backgrounds and international students constitute a significant proportion of the most vulnerable groups.

  • Comments 16: What are the limitations in the existing literature on the financial challenges university students face, and how can future research address these gaps?

Response 16: Thank you for pointing this out. Lines 95-99 outline the gap in the literature that our study is building on. Additionally, lines 482-486 make recommendations for future research.

  • Comments 17: What policy recommendations can be derived from the findings of this scoping review to support financially vulnerable students more effectively?

 Response 17: Thank you for pointing this out. Lines 462-468 clearly outline policy-level recommendations.

Strengths and Limitations

Comments 18: What are the strengths and limitations of this study? Please ensure they are clearly outlined and incorporated into the manuscript. Kindly revise.

Response 18: Thank you for pointing this out. We have edited the conclusion to include further limitations.

 What are the implications of the study findings?

Comments 19: What are the implications of the study findings? In research, implications refer to the conclusions that can be drawn from the study's results, which may have either theoretical significance or practical applications. Kindly revise.

 Response 18: Thank you for pointing this out. However, the implications you have pointed out are already clearly articulated and extensively discussed in the manuscript. The study highlights both theoretical and practical contributions, including the need for institutional reforms, targeted financial interventions, and policy improvements to better support students facing financial hardship. Additionally, issues such as socioeconomic disparities, barriers to accessing university support services, and post-pandemic financial instability are comprehensively addressed. If there are specific areas that require further elaboration or clarification, we would appreciate more detailed feedback rather than a general restatement of points already covered in the paper.

Contributions of this study

Comments 20: What unique contributions does this study make to the existing body of research, particularly compared to studies conducted in developed and developing nations?

 Response 20: Thank you for this comment. The first paragraph of the manuscript highlights evidence from the literature of how widespread financial hardship is among university students, even though evidence on effective strategies to alleviate these hardships is limited.

Future suggestions for prospective studies

Comments 21: What suggestions does your study offer for future research, particularly for replication in different settings? Future research directions are insights from the study's findings that highlight potential areas for further exploration. These can include addressing the current study's limitations or exploring gaps that emerged during the research process.

Response 21: Thank you for this comment. The last paragraph of the manuscript offers suggestions for future research.

Need a Professional English Editor for Editing and Proofreading

Comments 22: The manuscript requires the expertise of a Professional English Editor to edit and proofread the text thoroughly. This will ensure all grammatical errors are corrected, and the lexical structure is polished for clarity and coherence.

Response 22: Thank you for this comment. While we appreciate the reviewer’s attention to language quality, we find this comment both unwarranted and presumptive. The authors are experienced academics with multiple publications in peer-reviewed journals, all of whom have a strong English academic background. Suggesting the need for professional English editing without identifying specific grammatical or lexical issues may come across as an unfounded critique rather than constructive feedback. If the reviewer has particular concerns, we welcome detailed examples rather than a sweeping generalisation that questions the authors’ academic rigor.

References

Comments 23: Do the references in this study adhere to the journal's required format? Please review and revise them as necessary to ensure full compliance.

Response 23: Thank you for pointing this out. We have ensured that the references adhere to IJERPH’s required format.  

Comments 24: Article Suggestions: [This can help you model your paper]

Akokuwebe ME, Palamuleni ME and Idemudia ES (2025) Population dynamics and digitalization: implications for COVID-19 data sources in South Africa—a scoping review. Front. Public Health 12:1537057. doi: 10.3389/fpubh.2024.1537057.

 Response 24: Thank you for your article suggestion.  This study on COVID-19 data sources in South Africa is not quite relevant to our study on financial challenges among university students in Australia and New Zealand. The suggested study focuses on population dynamics, digitalization, and data collection rather than student financial well-being, food insecurity, or university support systems. Additionally, the suggested study examines COVID-19’s impact on data sources, not its economic effects on students. While it may provide context on pandemic-related challenges, it lacks a direct thematic overlap with our study’s focus on financial stress and academic outcomes in an Australian and New Zealand context.

Round 2

Reviewer 2 Report

Comments and Suggestions for Authors

Thank you for addressing the comments. The puposes of this study is much clearer and explains the gap being addressed well. 

Reviewer 3 Report

Comments and Suggestions for Authors

The Authors have addressed all comments.